# Recent Advances in Colorimetric Sensors Based on Gold Nanoparticles for Pathogen Detection

**DOI:** 10.3390/bios13010029

**Published:** 2022-12-26

**Authors:** Jianyu Yang, Xin Wang, Yuyang Sun, Bo Chen, Fangxin Hu, Chunxian Guo, Ting Yang

**Affiliations:** 1Institute of Materials Science and Devices, School of Materials Science and Engineering, Suzhou University of Science and Technology, Suzhou 215009, China; 2Research Center for Analytical Sciences, Department of Chemistry, College of Sciences, Northeastern University, Shenyang 110819, China

**Keywords:** gold nanoparticles, infectious pathogens, colorimetric sensor, point-of-care test

## Abstract

Infectious pathogens cause severe threats to public health due to their frightening infectivity and lethal capacity. Rapid and accurate detection of pathogens is of great significance for preventing their infection. Gold nanoparticles have drawn considerable attention in colorimetric biosensing during the past decades due to their unique physicochemical properties. Colorimetric diagnosis platforms based on functionalized AuNPs are emerging as a promising pathogen-analysis technique with the merits of high sensitivity, low-cost, and easy operation. This review summarizes the recent development in this field. We first introduce the significance of detecting pathogens and the characteristics of gold nanoparticles. Four types of colorimetric strategies, including the application of indirect target-mediated aggregation, chromogenic substrate-mediated catalytic activity, point-of-care testing (POCT) devices, and machine learning-assisted colorimetric sensor arrays, are systematically introduced. In particular, three biomolecule-functionalized AuNP-based colorimetric sensors are described in detail. Finally, we conclude by presenting our subjective views on the present challenges and some appropriate suggestions for future research directions of colorimetric sensors.

## 1. Introduction

The ongoing epidemic of COVID-19 has announced that infectious diseases are still one of the leading causes of death for people around the world [1,2]. With the change in modern lifestyle, infectious diseases can spread rapidly worldwide, posing a severe threat to human health and causing substantial economic loss [3]. In particular, pathogenic bacteria such as *Bacillus cholerae*, *Streptococcus pneumoniae*, *Mycobacterium tuberculosis*, and *Clostridium botulinum* can lead to severe human illnesses and enormous numbers of deaths per year [4]. Antibiotic overuse and the lag in the discovery of novel replacements exacerbate the situation and accelerate the formation of multidrug-resistant bacterial strains [5,6]. Further, viruses-borne infectious diseases are considered to be one of the most serious threats to human health [7]. For instance, the SARS virus, Zika virus, human immunodeficiency virus, and the recent outbreaks of SARS-CoV-2 virus and monkeypox virus have already caused thousands of deaths [8]. Hence, the rapid and accurate detection of pathogens (such as pathogenic bacteria, viruses, fungi, etc.) is critical not only for preventing the wider spread of infectious diseases and improving medical healthcare systems, but also for monitoring food safety and protecting environmental ecosystems [9,10]. Nowadays, a wide range of diagnosis strategies have been successfully developed for the detection of pathogenic bacteria, including plating and culturing [11], surface-enhanced Raman scattering (SERS) [12], mass spectrometry (MS) [13], and next-generation sequencing-based methods [14]. Nevertheless, these strategies are often confined by some demerits, such as being time-consuming, involving tedious procedures, necessitating sophisticated equipment, and/or requiring proficient operators. Similarly, antibody-based assays [15], polymerase chain reaction (PCR) [16], and quantitative real-time reverse-transcription polymerase chain reaction [17] have also been employed for virus detection in clinical diagnosis. However, these methods have difficultly fully meeting rapid and accurate point-of-care testing for infectious diseases. Recently, colorimetric sensors have been considered as an emerging strategy to revolutionize pathogen detection [18,19,20]. Compared to electrochemical sensors [21], the most attractive aspect of colorimetric systems is that the whole analysis process can be carried out only by the naked eye without the assistance of sophisticated instruments [22,23,24].

Gold nanoparticles (AuNPs), an intensively investigated nanomaterial, have been widely employed in biosensing, drug delivery, disease diagnosis, biolabeling, and other exciting fields due to their unique physical properties [25,26,27]. The size of AuNPs is approximately 1–100 nm, and they are usually colloidal when dispersed in an aqueous solution. Functionalized AuNPs have excellent stability and biocompatibility, which makes them suitable for analyte detection in complex biological systems. As a sensing unit, the intrinsic properties of AuNPs provide several advantages for the construction of colorimetric sensor [28,29]. Firstly, the most attractive aspect is that AuNPs exhibit strong surface absorption characteristics [30]. When the free electrons on the surface of AuNPs are excited by incident light, a surface plasmon resonance effect is generated. A characteristic resonance peak can be displayed in the ultraviolet–visible light region. The emergence of AuNPs can be regarded as a milestone in the construction of a colorimetric sensor [31]. Secondly, the synthesis and functionalization of AuNPs are very easy. By regulation of the morphology of AuNPs, the distance between particles, and the refractive index of surrounding media, AuNPs with different colors can be obtained easily. Due to their large surface-to-volume ratio, AuNPs can be easily functionalized by various ligands such as antibodies, peptides, aptamers, and other small biomolecules, which endows them with the ability to recognize analytes. Thirdly, AuNPs also exhibit excellent fluorescence-quenching performance and outstanding catalytic ability of the mimic enzyme. As an efficient fluorescence quencher, AuNPs have been widely utilized to construct Forster resonance-energy transfer (FRET) [32] and nanometal surface-energy transfer (NSET) systems [33] for highly sensitive detection of analytes. Forth, AuNPs have outstanding peroxidase-like catalysis activity. A colorimetric sensor based on the intrinsic peroxidase-like activity of AuNPs is realized by catalyzing the oxidation of chromogenic substrates such as 3,3,5,5-tetramethylbenzidine (TMB) [34]. Finally, colorimetric sensors based on AuNPs can be combined with the latest detection techniques, and advanced machine learning algorithms further significantly enhance the detection performance [35]. Hence, colorimetric sensors based on functionalized AuNPs have attracted growing interest in the field of biosensing [36,37,38]. Based on the corresponding color change of AuNPs as the analytical signal, these colorimetric biosensors have been employed for various types of pathogen detection [39,40,41]. In comparison to traditional pathogen detection methods, AuNPs-based sensor arrays have the characteristics of easy readout, portability, and cost-efficiency, providing potential ability for rapid and accurate point-of-care tests of pathogens. Therefore, a timely and comprehensive summary of this research field is essential.

In this review, we focus on recent advances in the development of colorimetric sensors based on the unique physical properties of AuNPs for pathogen detection. The strategies for colorimetric detection of pathogens in biological samples are introduced, including application of indirect target-mediated aggregation, the chromogenic substrate-mediated catalytic activity method, point-of-care testing (POCT) devices, and machine-learning-assisted colorimetric sensor arrays (Figure 1). The principles of pathogen recognition and colorimetric signal readout for pathogen detection are highlighted. In particular, various recognition-ligand-functionalized AuNPs, including antibodies, peptides, aptamers, and phages, are discussed in detail. The relative POCT applications such as lateral flow assays and microfluidics are introduced and evaluated. In addition, common machine learning algorithms involved in the colorimetric sensor array are viewed. Finally, we conclude by outlining challenges to and perspectives for colorimetric sensors.

## 2. Target-Mediated AuNP-Aggregation-Based Colorimetric Sensors

Localized surface plasmon resonance (LSPR) describes the phenomenon of resonance oscillation of free electrons on the surfaces of particles in the presence of light [42]. This phenomenon can be used to study the aggregation state of particles in solution based on the unique optical properties of AuNPs. The fundamental mechanism of the construction of colorimetric strategies based on AuNPs mainly relies on distance-dependent LSPR [43]. It is worth noting that interparticle plasmonic coupling could induce redshift of the UV–vis spectrum and lead to a color change of the system. The dispersed, spherical AuNPs exhibit wine-red color in solution, with the plasmonic band around 520 nm. The color of aggregated AuNPs changes from red to blue as the diameter gradually increases. However, the redispersion of agglomerated AuNPs can lead to color change from blue to red. The different states of AuNPs in solution can exhibit distinct colors. The aggregation state of AuNPs highly depends on the surface microenvironment, and changes in the external environment can induce aggregation or dispersion. Based on this simple principle, colorimetric sensors based on target-induced AuNP aggregation have become the most widely used strategy for analyte detection [44,45]. In addition, AuNPs can be efficiently functionalized by biomolecules and have excellent biocompatibility [29,46]. The most common recognition elements for bacterial cells include antibodies, peptides, aptamers, bacteriophages, and other small biomolecules [47,48]. Suitable employment of these recognition elements in the sensing system can drastically simplify the cost of testing, shorten the sample preparation time before analysis, and significantly improve detection performance. Next, we review various recognition-element-functionalized AuNPs in the construction of colorimetric sensors for pathogen detection.

### 2.1. Antibody-Functionalized Gold Nanoparticles

An antibody is a Y-shaped glycoprotein that can perform specific binding with an antigen and has various immunologic functions. The high affinity between antigens and their corresponding antibodies provides an excellent strategy for the construction of colorimetric sensors for bioanalyte detection. Covalent cross-linking (such as EDC-NHS chemistry) and physical adsorption (such as pH-mediated adsorption) are the two main methods to conjugate antibodies on the surfaces of AuNPs [49]. Significantly, antibody–AuNPs bioconjugates are widely employed in the field of pathogen detection. For example, Pedrosa et al. developed a simple colorimetric immune-sensing system for *Lactobacillus* species (spp.) and *Staphylococcus aureus* (*S. aureus*) detection with antibody-functionalized AuNPs [50]. The detection mechanism is based on the antibody–antigen recognition interaction between the antibody–AuNPs and the target bacteria, which further leads to the color change of the system upon AuNPs aggregation. The detection limits of *Lactobacillus* spp. and *S. aureus* are 105 and 120 CFU/mL, respectively. Singhal et al. presented a fast and specific biosensor for foodborne-bacteria detection based on graphene-oxide-coated AuNPs (GO–AuNPs) [51]. The recognition antibodies were covalently conjugated to the surface of GO–AuNPs by EDC-NHS chemistry (Figure 2A). In the presence of the target bacteria (*Escherichia coli* (*E. coli*) and *Salmonella typhimurium* (*S. typhimurium*)), GO–AuNPs would quickly aggregate and cause the color change within 5 min. The detection limit of the colorimetric method for *E. coli* is 10^3^ CFU/mL by the naked eye. The LOD can be reduced to 10^2^ CFU/mL by spectrophotometric techniques. In addition, antibody-functionalized GO–AuNPs can specifically inactivate the bacteria after NIR exposure. Hence, this method can realize rapid bacterial detection and the extermination of bacteria simultaneously, providing a new way to analyze pathogenic bacteria.

Similarly, antibody–AuNPs also show great potential in virus detection. Ray et al. constructed a novel sensing system based on anti-spike antibody-attached AuNPs for rapid SARS-CoV-2 virus diagnosis and inhibition [52]. Upon the addition of SARS-CoV-2 antigens or virus particles, the antibody–AuNPs can aggregate immediately and cause the solution’s color to change from pink to blue owing to the antibody–antigen interaction (Figure 2B). This sensing system can realize visual detection of the specific viral antigen or of pseudo-SARS-CoV-2 within 5 min. The limit detections of SARS-CoV-2 antigen and virus particles were 1 ng/mL and 1000 particles/mL, respectively. For further susceptibility detection, 4-amino thiophenol was modified on the surface of antibody–AuNPs to construct a SERS probe. The system could detect SARS-CoV-2 antigen even at a level of 4 pg/mL and virus particles at a concentration of 18 particles/mL. In recent studies, some researchers have also developed a sensing system for virus antibodies detection. For example, Su et al. designed and constructed a nano-sensor platform for rapid detection of SARS-CoV-2 antibody based on specific epitope–IgG interactions [53].

### 2.2. Peptide-Functionalized Gold Nanoparticles

Peptides are a new bio-functional candidate molecule that have played a significant role in the targeted recognition and detection of biochemical analytes in recent years [54]. Besides their natural physiological functions, peptides can be reasonably synthesized or screened from peptide libraries and can be employed to modify nanostructures to obtain effective recognition probes [55]. Significantly, peptide–AuNP-based colorimetric nano-platforms have drawn increasing interest in pathogen detection. For instance, Chung et al. employed a 12-mer specific binding peptide as the recognition element for the selective detection of fungal *Aspergillus niger* (*A. niger*) [56]. The *A. niger* binding peptide was obtained through phage-display screening. The binding peptide was immobilized on the surface of AuNPs by a Au-S coordination interaction. In the presence of *A. niger* spores, the peptide–AuNPs immediately aggregate, resulting in a visible color change of the solution (Figure 2C). This simple colorimetric system obtained accurate detection of *A. niger* even at a level of 50 CFU/mL within 10 min with the assistance of a smartphone-based image application. In a recent study, Liu et al. developed a colorimetric strategy for SARS-CoV-2 primary protease detection based on peptide–AuNPs [57]. The detection principle was that the peptide substrate of the main protease induces aggregation of AuNPs and causes a color change of the solution. Furthermore, the researchers also employed an electrochemical method for the main protease detection. The sensitivity and anti-interference ability of the sensing system has been further enhanced.

Bacterial toxins pose a massive threat to the environment and to human health [58]. Hence, it is highly urgent to develop a simple and rapid method for bacterial toxin detection. Recently, peptide–AuNPs have been used as a colorimetric probe for toxin detection due to their excellent optical properties. For example, Liedberg et al. developed a colorimetric method to detect botulinum neurotoxin serotype A light chain (BoLcA) based on biotinylated peptide-functionalized AuNPs [59]. The designed biotinylated peptides induce the aggregation of AuNPs. In addition, the peptides contain a BoLcA recognition site, so BoLcA can cut off the peptides to prevent AuNP aggregation. The researchers carefully designed and prepared two peptide substrates, and successfully realized BoLcA detection by the colorimetric sensing system. In another study, Li et al. presented a rapid and sensitive colorimetric assay for lipopolysaccharide (LPS) detection based on an LPS-binding peptide and AuNPs [60]. LPS, also known as bacterial endotoxin, is the main structural component of the outer member of gram-negative bacteria [61]. The designed peptide terminal contains a C-terminal cysteine thiol, which can be anchored on AuNPs through the Au-S bond. Meanwhile, the positively charged peptides decrease the negative charge density on the surface of AuNPs and eventually lead to aggregation. Upon addition of LPSs to the system, AuNPs changed from an aggregated state to a well-dispersed state due to the interaction between the peptides and LPSs. The limit detection of the colorimetric sensor for LPSs is 2.0 nM, with an excellent linear range from 10–1000 nM.

### 2.3. Aptamer-Functionalized Gold Nanoparticles

Aptamers are single-stranded oligonucleotides (DNA or RNA) and are usually obtained from the nucleic acid molecular library by in vitro selection. Aptamer-functionalized nanomaterials have been employed in applications ranging from biosensing to diagnostics due to their high binding affinity towards target analytes [62]. Aptamer-functionalized AuNPs have displayed enormous molecular promise for applications in the detection of various types of pathogens in recent studies [63]. For instance, Pei et al. developed a colorimetric platform for methicillin-resistant *S. aureus* (MRSA) detection based on enzyme-driven stochastic DNA walkers [64]. The two types of released multiple-walking DNA chains could continuously walk on the AuNPs based on a 3D track upon the addition of MRSA (Figure 2D). Then, the state of DNA–AuNPs was unstable, and they aggregated, resulting in the color of the system changing from red to blue, which can serve as the colorimetric signal for MRSA detection. The system displayed an excellent linear response from 10^0^ to 10^5^ CFU/mL with the LOD at 1 CFU/mL. In addition, this colorimetric biosensor could realize accurate detection of MRSA in human body fluids. Deng et al. presented a colorimetric method for *E. coli K88* (*ETEC K88*) detection based on a G-quadruplex and DNA-functionalized gold nanoparticles (AuNPs) [65]. The AuNPs were firstly modified by capturing DNA. The aptamer formed a stable double-stranded structure with the capture of DNA–AuNPs. With the addition of target bacteria, the double strands of captured DNA and aptamer were untied due to the tighter binding interaction between bacteria and aptamer. Then, the G-quadruplex combined with the captured DNA–AuNPs and resulted in the solution changing from red to blue. This colorimetric sensor showed a good linear response 10^2^ to 10^6^ CFU/mL, with the LOD at 1.35 × 10^2^ CFU/mL.

DNA–AuNPs have also been widely utilized in virus detection. For example, Aithal et al. employed aptamer–AuNPs as colorimetric probes for early diagnosis of COVID-19 [39]. The aptamer–AuNPs could specifically bind to the spike protein on the SARS-CoV-2 membrane. With the assistance of a coagulant (MgCl_2_ salt), the AuNPs aggerated and changed the color of the solution. The LOD of the system was 16 nM in phosphate-buffered saline by plasmon absorbance spectra. CRISPR/Cas systems are widely utilized to improve the sensitivity of molecular detection. For accurate SARS-CoV-2 detection, Zhou et al. constructed a magnetic pull-down-assisted colorimetric strategy based on a CRISPR/Cas12a system. SARS-CoV-2 could be distinguished from other similar, related viruses with the assistance of the CRISPR/Cas12a system [66]. This colorimetric method could realize accurate SARS-CoV-2 RNA detection based on DNA–AuNPs by the naked eye. The LOD of this method was 50 RNA copies per reaction under optimal conditions.

**Figure 2 biosensors-13-00029-f002:**
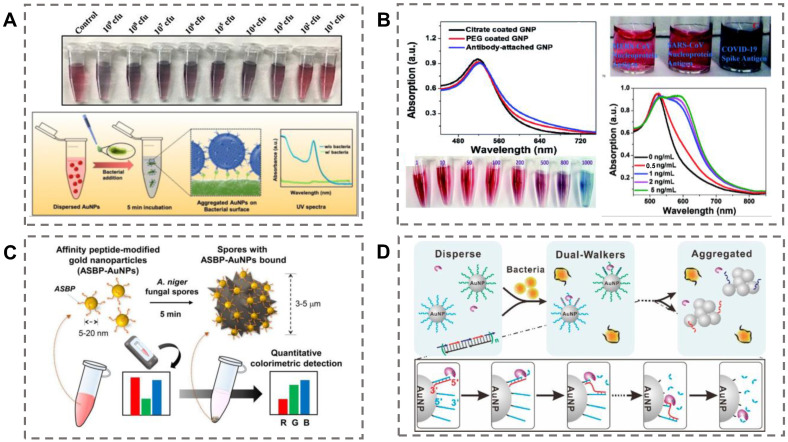
Colorimetric detection of pathogens based on antibody-, peptide-, and aptamer-functionalized AuNPs. (**A**) A colorimetric sensor with antibody-conjugated PEG-GO–AuNPs for *E. coli* and *S. typhimurium* detection. Reproduced with permission [51]. Copyright 2021, Elsevier. (**B**) A colorimetric sensor with antibody-attached AuNPs for rapid SARS-CoV-2 virus detection. Reproduced with permission [52]. Copyright 2021, Royal Society of Chemistry. (**C**) A colorimetric sensor with peptide-attached AuNPs for allergenic fungal spore detection. Reproduced with permission [56]. Copyright 2021, Elsevier. (**D**) A colorimetric sensor with aptamer-modified AuNPs for MRSA detection. Reproduced with permission [64]. Copyright 2020, American Chemical Society.

### 2.4. Other Biomolecule-Functionalized Gold Nanoparticles

Vancomycin (Van) is a glycopeptide antibiotic that is widely used in the detection of gram-positive bacteria due to its ability to bind to D-Ala-D-Ala on the cell walls specifically [67]. Considering this, Chen et al. constructed a visual colorimetric method for Gram-positive bacteria detection [68] (Figure 3A). The common antibiotic Van was chosen as the ligand to synthesize AuNPs by a simple one-pot reaction. The obtained Van–AuNPs specifically combined with Gram-positive bacteria. This phenomenon not only distinguishes between Gram-positive and Gram-negative bacteria but can also be employed for accurate quantitative analysis of gram-positive bacteria. The LODs of this method based on bare eyes for three types of Gram-positive bacteria, including *S. aureus*, *M. luteus*, and *B. subtilis*, were 1 × 10^9^, 1 × 10^9^, and 1 × 10^9^ cells/mL, respectively. In addition, the system realized the accurate detection of Gram-positive bacteria in tap water and orange juice.

D-amino acids are involved in the formation of peptidoglycans in the bacterial wall, while the metabolism of animal cells only uses L-amino acids. Therefore, using D-amino acids and their derivatives can effectively distinguish bacteria and animal cells and selectively label bacteria [69]. For instance, Jiang et al. utilized D-amino-acid-modified AuNPs as a colorimetric probe for pathogen visual detection in the clinic [70]. The researchers selected sodium borohydride as the reducing agent and D-amino acid as a template to prepare AuNPs with uniform size and stable optical properties (Figure 3B). The D-amino acid–AuNPs aggregated on the surface of bacteria, and the color of the solution changed from red to blue upon the addition of target bacteria. *S. aureus* and MRSA could be clearly distinguished based on the difference in the color change of the system. It is worth noting that the colorimetric assay realized bacteria detection in ascites samples from patients.

Phages are viruses that attack bacteria and contain nucleic acids and proteins. Phages have been employed as a recognition element for pathogen detection due to their low production cost and stable storage conditions [71]. Phage-protein-functionalized AuNPs can capture the target analyte, changing the color of the solution. In a recent study, Chen et al. combined the optical property of AuNPs with chimeric phages targeting bacteria to construct a colorimetric system [72]. The capsids of chimeric phages were thiolated by the EDC-NHS reaction. The phages could bind to the target bacteria due to the interaction between receptor-binding proteins and the receptors on bacteria (Figure 3C). Then, the AuNPs aggerated on the phage surface due to the Au-thiol interaction and generated the corresponding colorimetric signal. The LOD of this assay is 100 CFU/mL by the naked eye.

**Figure 3 biosensors-13-00029-f003:**
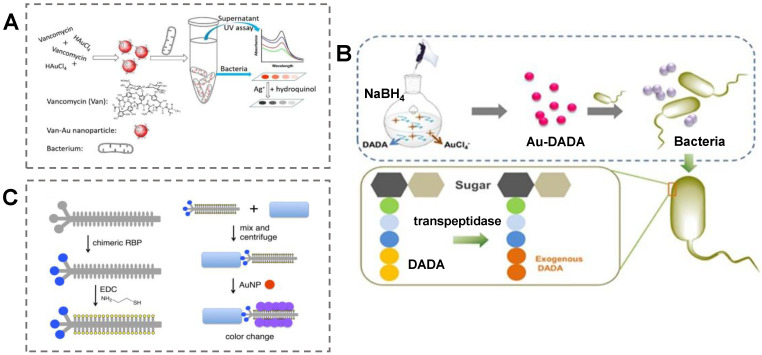
Colorimetric detection of pathogens based on small-molecule-functionalized AuNPs. (**A**) A colorimetric assay using vancomycin-decorated Au nanoparticles for bacteria detection. Reproduced with permission [68]. Copyright 2019, Elsevier. (**B**) A colorimetric assay with D-amino acid- AuNPs for bacteria detection. Reproduced with permission [70]. Copyright 2018, Ivyspring International. (**C**) A colorimetric sensor with antibody-attached AuNPs for rapid SARS-CoV-2 virus detection. Reproduced with permission [72]. Copyright 2019, American Chemical Society.

In short, various types of recognition-element-modified AuNPs have been developed as colorimetric probes for pathogen detection, and further studies are required to develop better applications. First, it is significant to investigate efficient functionalization strategies for AuNPs with high stability and homogeneity. In addition, colorimetric sensors based on AuNPs should not only realize accurate detection of pathogens in solution, but also need to further explore point-of-care testing of pathogens in complex biological samples by other devices, such as paper-based devices and microfluidic systems.

## 3. Chromogenic Substrate-Mediated Catalytic-Activity-Based Colorimetric Sensors

In addition to the LSPR effect, the electric activity on the surface of AuNPs also endows them with the property of mimicking peroxidase. As an artificial enzyme, AuNPs are expected to overcome the shortcomings of biological and traditional artificial enzymes in colorimetric strategies. In recent years, colorimetric sensors based on the inherent catalytic properties of AuNPs have been widely employed in bioanalyte detection to enhance the sensitivity and selectivity of sensing system [73]. The enzyme-like activity of AuNPs is closely related to their morphology, aggregation state, and the surface ligand. With the assistance of chromogenic substrates, colorimetric sensors based on the peroxidase-mimicking activity of AuNPs can be employed to detect pathogens. Chromogenic substrates such as 4-nitrophenol (4-NP), 3,3′,5,5′-tetramethylbenzidine (TMB), 2,2′-azino-bis (3-ethylbenzothiazoline-6-sulfonic acid) diammonium salt (ABTS), and ortho-phenylenediamine (OPD) are commonly chosen as the colorimetric output indicators [74].

In a recent study, Zhao et al. prepared a novel nanoconjugate composed of antibody-functionalized Fe_3_O_4_/Au magnetic nanoparticles and aptamer–AuNPs as a colorimetric probe for *S. aureus* detection [75]. The antibody-modified magnetic nanoparticles selectively captured and rapidly separated the target bacteria from complex matrices. Additionally, anti-*S. aureus* aptamer-modified AuNPs were used as the artificial enzyme to catalyze TMB to obtain the optical signal. The absorbance of the solution decreased linearly with increasing concentrations of *S. aureus* ranging from 10 to 10^6^ CFU/mL. The LOD for *S. aureus* was 10 CFU/mL at the optimal conditions.

Introducing other noble metals into AuNPs is an effective strategy to improve the catalytic performance. For instance, the Au@Pd nanoparticles and Au@Pt nanoparticles exhibited excellent peroxidase activity. Recently, Rezayan et al. presented a label-free colorimetric sensor based on the peroxidase-like activity of aptamer-modified Au@Pd nanoparticles for *Campylobacter jejuni* (*C. jejuni*) detection [76]. The recognition mechanism is based on the specific interaction between the aptamer and the surface protein in the cell membranes of *C. jejuni*. The selected aptamer covered the surface of Au@Pd nanoparticles by electrostatic interactions and led to the reduction of the catalytic activity of nanoparticles. In the presence of the target analyte, the aptamer detached from the surface of the Au@Pd nanoparticles and adsorbed to the bacteria. Hence, the catalytic properties of Au@Pd nanoparticles recovered and oxidized the chromogenic substrate TMB into dark blue. The absorbance of the system displayed a linear relationship with the concentration of *C. jejuni*, and the LOD was as low as 100 CFU/mL in milk. Further, 4-mercaptophenylboronic acid-functioned Au@Pt nanoparticles were constructed as a colorimetric probe for *E. coli O157:H7* detection [77]. The Au@Pt nanoparticles could bind to the surface of bacteria through the boronate–diol interaction and electrostatic adsorption. Based on the catalytic oxidation of the peroxidase substrate TMB, *E. coli O157:H7* could be detected with the naked eye. The LOD of the system was 7 CFU/mL, and the total detection time only took 40 min.

From the above examples, colorimetric sensors based on the peroxidase-mimicking property of AuNPs exhibit promising application prospects in pathogen detection, but some drawbacks are also unavoidable. Primarily, it is of great significance to develop a new synthesis strategy for preparing AuNPs with stable performance and uniform size in large quantities. The morphology of the nanoparticles displays a great influence on its catalytic performance. In addition, commonly interfering substances in biological samples, such as proteins, carbohydrate, and ions, may inhibit the catalytic property of AuNPs. The anti-interference ability of the sensing system needs to be further improved to realize pathogen detection in actual samples such as human urine and oral flora. Finally, other types of chromogenic agents should be explored to improve the stability and sensitivity of the detection system.

## 4. Point-of-Care-Testing Colorimetric Sensors Based on AuNPs

Compared with the highly precise diagnostic testing and analysis by complex instruments, the colorimetric sensing strategy displays a vast application prospect in resource-limited regions. The target analytes can be accurately identified based on the sensitive color change of the system as the output signal. The advantage of this method is that the analysis process can be realized only by the naked eye without the assistance of sophisticated instruments. Hence, the colorimetric strategy is suitable for field analysis and point-of-care testing (POCT) [78]. Recently, various types of POCT diagnostic devices have been designed and presented to reduce the traditional complicated detection operations [9,79,80]. POCT equipment mainly includes a sensing system and a readout system. POCT devices should be prepared to be simple and to be operated by non-professionals. With the assistance of gold nanoparticles, precise results can be obtained with the naked eye and/or a smartphone application. In the following, we briefly introduce two types of colorimetric POCT devices for bacteria analysis and detection.

### 4.1. Lateral-Flow-Assay-Based Colorimetric Sensor

A lateral flow assay (LFA) device was initially termed by Leuvering et al. [81]. LFA devices have been employed for various target detection, such as cancer marker proteins in the blood and pathogens in the urine [82]. AuNPs are the most widely used colorimetric probe to construct LFA devices due to their simple synthesis, low cost, and excellent biocompatibility. Meanwhile, AuNPs can be functionalized with recognition elements by surface passivation using electrostatically charged antibodies or thiol-modified aptamers. An LFA device typically employs antibodies as the recognition element to combine with the target analyte [83].

LFA devices are typically constructed from the mobile phase and the stationary phase (Figure 4A). Once the analyte attaches to the sample pad, the analyte is driven to pass through the sample pad through the capillary force generated. Then, the sample reaches the conjugate pad. If the sample contains the target, it interacts with the antibody-functionalized AuNPs. Next, the sample passes into the nitrocellulose membrane, where a control line (C-line) and a test line (T-line) exist. The C-line contains the antibodies that can selectively bind to the primary antibody in the conjugate pad. If the C-line displays the wine-red color of AuNPs, the test is valid. Further, the T-line consists of antibodies that specifically bind to the target analyte. A red line can be formed if the analyte is “sandwiched” between antibodies immobilized in the T-line and antibodies immobilized on the AuNPs.

There are numerous examples of LFA devices for bacteria detection [84,85]. For instance, Ju et al. prepared p-mercaptophenylboronic acid-modified AuNPs (PMBA–AuNPs) as the colorimetric probe to construct a new capture-antibody-independent LFA method for bacteria detection [86]. The PMBA–AuNPs displayed outstanding capability for Gram-negative and Gram-positive bacteria detection by covalent bonding (Figure 4B). The sensing system showed an excellent linear range from 10^3^–10^7^ CFU/mL for *E. coli O157:H7* based on the gray value of the T-line by the naked eye. Wang et al. constructed a label-free LFA system for bacteria detection by introducing positively charged AuNPs ((+)AuNPs) [87]. Since the surfaces of bacteria are negatively charged, the (+)AuNPs could bind to bacteria due to the electrostatic interactions. Then, the (+)AuNPs-bacteria composite was specifically captured by a monoclonal antibody (McAb) immobilized on the T-line (Figure 4C). The LOD of the proposed LFA device was 103 CFU/mL for *Salmonella enteritidis*. In addition, the system achieved outstanding results in the detection of bacteria in drinking water, lettuce, and pork samples.

However, these two examples might give false negative results in practical samples due to the absence of the C-line. In another study, Ji et al. presented a point-of-care LFA device for the quantitative detection of *S. Enteritidis* by streptavidin-coated gold nanoparticles [88]. The prepared AuNPs were bound tightly to DNA by end-modified biotin. A smartphone was employed to collect the image information of the LFA strip for quantitative analyte analysis. The whole detection process only took 40 min and had a detection limit of 91.4 CFU/mL for *S. Enteritidis* samples. LFA devices have also been widely utilized in virus detection. Zhou et al. presented a CRISPR/Cas9-mediated triple-line LFA device combined with multiplex reverse transcription-recombinase polymerase amplification for dual-gene SARS-CoV-2 rapid detection in a single-strip test [89]. This dual-gene detection system significantly improved the accuracy of the sensing system for SARS-CoV-2. This newly developed device output four potential test results. Visibility of the C-line represented the negative result. Visibility of the T1 line and C line or T2 line and C line displayed suspected infection. Visibility of all three lines showed the patient was infected.

**Figure 4 biosensors-13-00029-f004:**
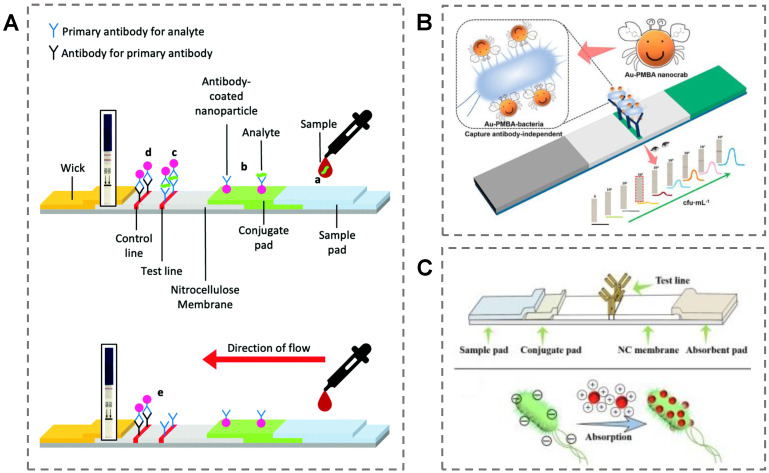
(**A**) Schematic of a prototypical LFA device. The device constituents in a successful positive test and a successful negative test. Reproduced with permission [82]. Copyright 2022, Royal Society of Chemistry. (**B**) A capture-antibody-independent LFA strategy based on PMBA–AuNPs for pathogen detection. Reproduced with permission [86]. Copyright 2022, American Chemical Society. (**C**) A label-free immunoassay LFA device with (+)AuNPs for *S. enteritidis* detection. Reproduced with permission [87]. Copyright 2019, Elsevier.

In addition to LFA devices, other types of paper-based colorimetric sensors have also been developed for pathogen detection [90]. For example, Kim et al. presented a novel paper-based radial flow chromatographic immunoassay with AuNPs as the chromatic agent for *E. coli O157:H7* detection [91]. Compared with the traditional antibody-function-modification method, this sensing platform chose the four-repeated gold-binding peptide-tagged (4GBP) streptococcal protein G (SPG) fusion protein as the bifunctional linker to immobilize antibodies on the surface of AuNPs. The prepared AuNPs displayed outstanding colloidal stability at high salt concentrations. The paper-based device pattern can be converted into grayscale values with the assistance of correlation-image-analysis software. The detection limit of this platform for *E. coli O157:H7* in whole milk was 10^3^ CFU/mL. In another similar study, Merkoci et al. constructed a novel colorimetric nano platform for *E. coli O157:H7* detection [92]. This sensing strategy selected AuNCs as the energy donor and antibody-functionalized AuNPs as the energy acceptor. The selected glass membranes were cut into 15 cm × 8 mm strips, and the mixture solution of antibody–AuNPs and AuNCs was added onto separate glass fiber strips. The assay was performed in a microtube. With the assistance of a smartphone, the biosensor achieved accurate bacteria detection with a limit of detection of 4.0 CFU/mL.

Although colorimetric sensors performed on a flat substrate format have many advantages over those in solution, there are still some technical drawbacks that need to be addressed. For the detection of pathogens in biological samples containing high concentrations of proteins, the protein corona formed on the surface of AuNPs may prevent binding with the target analyte, and the test may give false-negative results. In addition, the capture-antibody-modified AuNPs on the T-line may non-specifically bind to non-target analytes, resulting in a false positive. The false results may be removed by adding sucrose or tween or by nitrocellulose pretreatment. It is believed that more types of paper-based colorimetric devices will be designed for detection of pathogens shortly.

### 4.2. Microfluidic-Chip-Based Colorimetric Sensor

A microfluidic chip integrates various types of units and offers advantages such as a miniaturized analytical platform, high throughput, shortened detection time, and low sample consumption [93]. Significantly, the combination of AuNPs with a microfluidic chip can further improve the performance of detection system [94]. Considering the above advantages, microfluidic chips based on the opticality of AuNPs have been frequently reported for pathogen detection in recent years [95]. For instance, Lin et al. constructed a colorimetric microfluidic chip for *E. coli O157:H7* detection based on the color change by the aggregation of the AuNPs [96]. A smartphone was used to record the color change of the sensing system. The magnetic nanoparticles were functionalized by the capture antibodies, and the polystyrene microspheres were modified by the detection antibodies (Figure 5A). In the first mixing channel of the microfluidic chip, a magnetic-nanoparticle–*E. coli*–polystyrene complex was formed and captured in the separation chamber. After the addition of hydrogen peroxide into the complex, the AuNPs and the crosslinking agents reacted with the catalyst in the second mixing channel. Aggregation of the AuNPs was induced by the crosslinking of phenolic hydroxyl moieties and led to the system’s color changing from blue to red. The detection limit of this system for *E. coli O157:H7* was 50 CFU/mL. Recently, Stanciu et al. presented a paper-based microfluidic device for the simultaneous detection of *E. coli O157:H7* and *S. Typhimurium* with the quantification of the colorimetric signal [97]. Paper-based microfluidic devices are rapid, versatile, and low-cost. In this work, aptamer–AuNPs were decorated on the surface of polystyrene microparticles to enhance the stability and sensitivity of the colorimetric system (Figure 5B). The color intensity obtained by the image analysis equipment could be converted into the concentration of the analyte to achieve quantitative analysis. The LODs for *E. coli* O157:H7 and *S. Typhimurium* were 10^3^ CFU/mL and 10^2^ CFU/mL, respectively. In another study, Pan et al. presented a colorimetric microfluidic biosensor based on thiolated polystyrene microspheres (SH-PSs), which induced the aggregation of AuNPs for *S. typhimurium* detection [98]. The aptamer-functionalized PS–cysteamine conjugates were employed to react with bacteria, and the complementary-DNA-modified-magnetic-nanoparticle complex was used as the capture probe (Figure 5C). Upon the addition of *S. typhimurium* to the system, AuNPs aggregated on the surface of bacteria–aptamer–PS–cysteamine conjugates, causing the detection chamber to change color from red to blue. The microfluidic device displayed a reasonable specificity with an LOD of 6.0 × 10^1^ CFU/mL. To further improve the detection performance of the system, it is an excellent strategy to combine a DNA logic system with microfluidic technology [99].

The above examples demonstrated that integrating AuNPs into a microfluidic chip can significantly improve the detection performance of the system. Further development should be aimed at simplifying the experimental analysis process. In the future, it will be a trend to combine nanotechnology, photoelectric sensor technology, and 3D-printed technology to develop various types of POCT devices for the rapid detection of pathogens in daily life.

## 5. Machine-Learning-Assisted Colorimetric Sensor Arrays Based on AuNPs

Currently, sensor arrays have attracted much attention in pathogen detection due to their excellent sensitivity, simultaneous detection of multiple analytes, and easy operation [100]. The key to constructing a sensor array is the design and preparation of the sensor unit. Functionalized AuNPs are widely utilized in sensor arrays because of their excellent optical properties [101]. By modifying the surface of AuNPs, they can be endowed with the ability of target recognition [102]. AuNPs have been used to detect and kill bacteria, but it is still challenging to realize specific bacterial identification [103]. Currently, methods of targeting bacteria mainly rely on cationic polymers, antibodies, and carbohydrates [104]. In this section, we review the interaction mechanism between the functionalized AuNPs and the bacteria.

Different from the traditional “one probe vs. one analyte” sensor system, sensor arrays are composed of multiple sensing probes for the simultaneous identification of various analytes [105]. Simultaneous analysis of analytes is more conducive to studying the actual state of the analytes [106]. The obtained experimental data can be further analyzed by machine learning algorithms [107]. Machine learning can be divided into supervised or unsupervised learning algorithms according to whether the training data are labeled [35]. The training method with an objective is a supervised learning algorithm, and the training method without an objective is called an unsupervised learning algorithm. Next, we introduce these two types of machine learning algorithm used in AuNP-based sensor arrays.

### 5.1. Unsupervised-Algorithm-Based Sensor Array

An unsupervised learning algorithm has two primary functions in data analysis. The first is dimensionality reduction for extracting the representative characteristics from the high-dimensional data. The second is clustering analysis of data based on the pair-likeness measure. Standard unsupervised learning algorithms in data analysis include principal component analysis (PCA) [108], hierarchical cluster analysis (HCA), T-distributed stochastic neighbor embedding (T-SNE), and uniform manifold approximation and projection (UMAP) [109]. We first introduce the application of PCA in array-based sensors for bacteria identification. Recently, Yang et al. employed a AuNPs–Gold/silver nanocluster (AuAgNC) composite as the sensor unit to construct a three-dimensional sensor array for sulfur-containing species and bacteria discrimination [110]. The negatively charged AuAgNCs were bound to the surface of AuNPs by electrostatic interaction. The target analyte could induce the separation of AuAgNCs from AuNPs by the coordination interactions between sulfur-containing species and the surface ligand of AuAgNCs. (Figure 6A). Various target bacteria exhibited diverse affinities towards AuAgNCs due to their different surface microenvironments, leading to the color change of the solution. With the assistance of PCA, the obtained optical signals, including the fluorescence, UV–vis signal, and light scattering, could be reduced into a 2D plot. Five types of bacteria, including three sulfur-oxidizing bacteria and two common pathogens, could be well discriminated at a concentration of OD_600_ = 0.005. The HCA method can be used for visual analysis of analytes based on the similarity between distinct categories of data points. In another similar study, vancomycin-modified AuNCs (Van–AuNCs) and cetyltrimethylammonium-bromide-functionalized gold nanoparticles (CTAB–AuNPs) were selected as the optical probes to construct a sensor array for Gram-negative bacteria identification [111]. The recognition mechanism was based on the specific interaction between the vancomycin on the surface of AuNCs and the bacterial cell wall. In addition, the electrostatic interaction between the CTAB–AuNPs and bacteria also played an important role in bacterial identification. The negatively charged AuNCs could be absorbed on the surface of AuNPs to form a novel nanocomposite by electrostatic interaction. In the presence of target bacteria, three types of optical signals (fluorescence, UV–vis absorbance, and light scattering) were simultaneously changed due to the interaction between the bacteria and the nanocomposite. As shown in Figure 6B, ten types of Gram-negative bacteria at OD_600_ = 0.015 could be accurately identified by HCA based on the similarity of the different bacteria strains. Significantly, the sensing system accurately discriminated five subtypes of *E. coli*, including three types of antibiotic-resistant *E. coli* and two non-resistant *E. coli* strains. This study demonstrated that HCA can be used to evaluate the similarity of characteristics among diverse strains. As two emerging dimensionality reduction techniques, T-SNE and UMAP have more powerful functions in dealing with nonlinear problem. Unfortunately, these two methods are rarely utilized in colorimetric sensor arrays.

The above studies demonstrate the effectiveness of an unsupervised learning algorithms in complex data analysis from array-based sensors. However, unsupervised algorithms also have some disadvantages. Unsupervised algorithms are usually used for evaluating the initial dataset and cannot predict unknown analytes. In array-based sensor data analysis, we can use an unsupervised algorithm to extract the significant features from the sample data, serving for the subsequent supervised algorithm to provide accurate identification of microorganisms [112]. Next, we summarize supervised algorithms in the colorimetric sensor array.

### 5.2. Supervised-Algorithm-Based Sensor Array

Supervised learning algorithms are widely used to construct prediction models for pathogen identification [113]. They have two primary functions in data analysis: one is to solve the classification problem of unknown samples, and the other is to solve the regression problem. We can also understand that the classification problem corresponds to the qualitative analysis, while the regression problem is the qualitative analysis of the sample. LDA is the most-employed supervised algorithm for pathogen discrimination. LDA can reduce the dimensionality of sample data and retain the identification ability in the sensor array [114,115,116]. For instance, Wu et al. employed four different diverse charged surfaces of AuNPs as the colorimetric probe to construct a sensor array for pathogen identification [117]. Mercaptopropionic acid, mercaptosuccinic acid, cysteamine, and cetyltrimethylammonium bromide were chose as the surface ligands to obtain AuNPs with diverse surface charges. Upon the addition of microorganisms into the system, the rapid aggregation of AuNPs could induce the system color to change from red to blue. Various microorganisms displayed differential bindings to the AuNPs, providing a basis for microorganism identification. With the assistance of LDA, the collected four-dimensional data could be reduced into a 2D plot. As shown in Figure 7A, fifteen microorganisms, including twelve bacteria and three fungi, could be discriminated at a concentration of OD_600_ = 0.05. In a recent study, Liu et al. selected three types of D-amino acid (D-alanine, D-2,3-diaminopropionic acid, and D-glutamate)-functionalized AuNPs as the sensing elements to construct a colorimetric sensor array for microorganism identification [118]. The recognition mechanism is based on the fact that the D-amino acid–AuNPs could be incorporated into bacterial peptidoglycan and result in the color change of the sensing system (Figure 7B). Different types of bacteria exhibited diverse metabolic abilities of the D-amino acids, and the amount of AuNPs that accumulate on the surface of the bacteria was also different. LDA was employed to reduce the dimensionality of colorimetric signals in the 2D plot. Based on the diverse patterns of colorimetric signals, eight types of bacteria, even the antibiotic-resistant bacteria strains of *E. coli*, could be successfully discriminated. In addition, the colorimetric sensor array could also determine the minimum inhibitory concentration of antibiotics based on the metabolic activity of bacteria to D-amino acid–AuNPs. Random forest is also a typical representative of an unsupervised learning algorithm and has the advantages of fast training speed and strong generalization ability. It is especially suitable for microbial identification in the sensor array. For example, Xianyu et al. employed a random forest (RF) algorithm to accurately identify 19 types of microorganisms at multiple classification levels [119]. The biosynthesis of AuNP strategy has been regarded as a sensor array for microbial discrimination. Various features of the synthesized AuNPs, including the surface plasmon resonance (SPR) spectrum, the particle size, and the surface potential, have been utilized in the training data for establishing a mathematic model. About 80% of the measurements of biosynthetic AuNPs were used to construct the training set, and 20% of the measurements were for the test set. The accuracy of the testing results was nearly 100% for microbial identification. This example indicates that RF can significantly improve the recognition ability of array-based sensors for analyte identification. In addition, colorimetric sensor arrays based on AgNPs have also been reported for the identification of pathogenic strains [120,121].

With the rapid development of machine learning, more advanced supervised algorithms should be selected to enhance the quality of sample data in the sensor array. It is believed that colorimetric array sensors based on machine learning can play a great role in the early diagnosis of infectious diseases.

## 6. Conclusions and Overlook

As outlined in this review, various types of colorimetric strategies based on functionalized AuNPs have been widely utilized for pathogen detection, and the key characteristics of each sample, such as analyte, functionalization, strategy and LOD, are summarized in Table 1. Despite the significant progress in the construction of colorimetric systems based on AuNPs and their applications in pathogen detection, there are still numerous challenges and drawbacks in this promising field.

First, the sensitivity and selectivity of colorimetric sensor need to be improved to realize the detection of pathogens in actual samples, such as gut microbiota and oral flora. The morphology of AuNPs needs to be further optimized to improve the sensitivity of colorimetric pathogenic diagnostics. In general, decreasing the size of AuNPs can increase the specific surface area of AuNPs, leading to the enhancement of the peroxidase-mimicking property. In addition, the construction of a dual-signal output method combined with other analytical strategies such as electrochemistry, fluorescence, and SERS is also a way to improve the accuracy. The pattern of dual-signal output also greatly reduces the occurrence of false positive results. In addition, researchers should pay more attention to public health issues to develop novel colorimetric sensors for the detection of novel infectious diseases such as monkeypox virus [122].

Second, the current colorimetric-sensor-based AuNPs tend to detect only one type of pathogen and cannot detect multiple pathogens simultaneously. Although the emergence of machine-learning-assisted sensor arrays can enable the identification of pathogen species, the quantitative analysis of concentrations is rarely analyzed. Therefore, the performance of colorimetric sensors for multiple analyte detection should be further improved to achieve the simultaneous qualitative and quantitative analysis of multiple analytes. With the rapid development of artificial intelligence technology, the latest machine learning techniques, such as convolutional neural networks, also need to be integrated with array-based sensors to achieve accurate analysis of pathogens. The combination of colorimetric systems with multichannel microfluidic systems may be a promising strategy to improve multiple-target-detection abilities. In addition, the emergence of multi-drug resistance has caused extensive morbidity and mortality worldwide [123,124]. The development of an integrated platform based on AuNPs for the detection and inactivation of multi-drug resistant strains is expected to solve this problem [125]. Third, tremendous efforts should be paid to the practicability and commercialization of colorimetric diagnostic systems. One of the significant challenges is how to reduce the cost of functionalized AuNP preparation. The conjugation of recognition elements on the surface of AuNPs is usually time-consuming and labor-intensive. Therefore, the integration of AuNP preparation and modification of recognition elements into one step can significantly reduce the time and costs for the construction of colorimetric sensors. Further, colorimetric-sensor-based AuNPs can be further combined with emerging portable devices such as paper-based chips to achieve portability and intellectualization in clinical applications. With the assistance of a smartphone readout application, visualization and in situ detection of bacteria can be achieved. Therefore, the synergetic integration of the technologies discussed above can promote practical application of colorimetric sensors in pathogen detection.

In summary, we introduce the recent advances in the development of colorimetric sensors based on the unique optical and catalytic properties of AuNPs for pathogen detection. AuNPs are the most widely employed nanomaterials for visible signal readout in analytical methods. As mentioned above, an accurate colorimetric pathogen diagnostics platform can be improved by employing functionalized AuNPs as signal transduction and amplification tools. Overall, it is believed that this timely review can help researchers better understand the recent progress in colorimetric sensors based on AuNPs for pathogen detection. It is expected that this comprehensive review can also promote the rational design and construction of rapid, portable, low-cost, selective, and sensitive colorimetric sensors for a wide range of applications.

## Figures and Tables

**Figure 1 biosensors-13-00029-f001:**
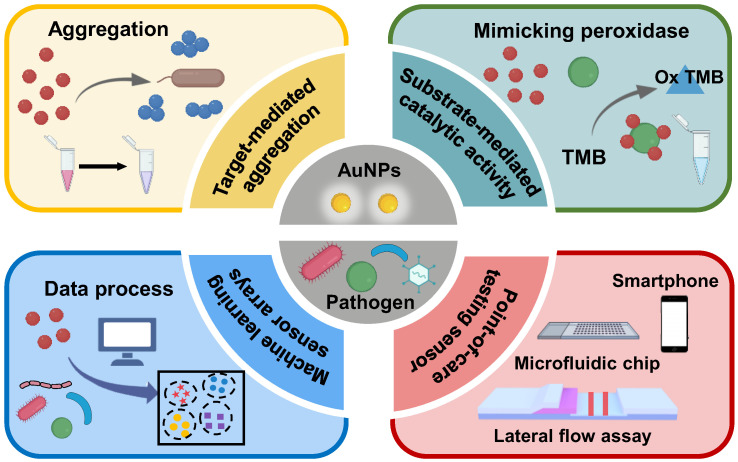
Schematic representation of colorimetric sensors for pathogen identification based on functionalized AuNPs.

**Figure 5 biosensors-13-00029-f005:**
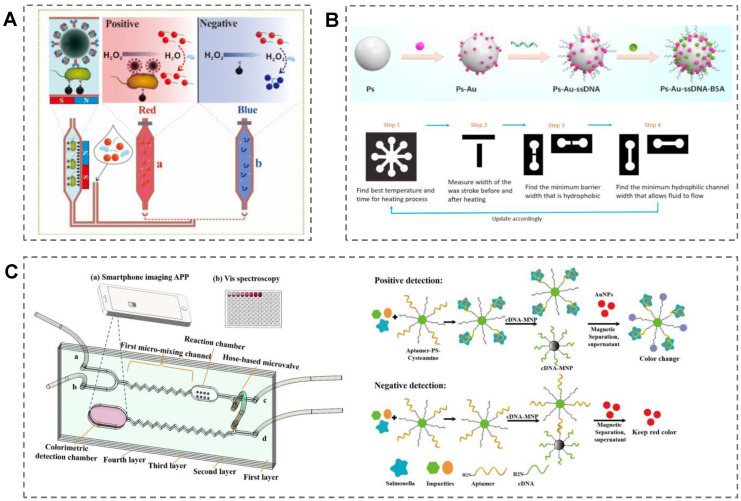
Colorimetric microfluidic chips based on functionalized AuNPs for pathogen detection. (**A**) A microfluidic colorimetric biosensor based on AuNP aggregation for *E. coli* O157:H7 detection. Reproduced with permission [96]. Copyright 2019, Elsevier. (**B**) A novel paper-based colorimetric microfluidic device for the simultaneous detection of *E. coli* O157:H7 and *S. Typhimurium* based on PS–Au–ssDNA–BSA microparticles. The 3 D structural of the designed microfluidic chip is shown on the left of figure. The specific recognition mechanism based on AuNPs is shown on the right of figure. Reproduced with permission [97]. Copyright 2022, Elsevier. (**C**) A microfluidic colorimetric strategy for the detection of *S. typhimurium* by using SH–PSs to aggregate AuNPs. Reproduced with permission [98]. Copyright 2021, Elsevier.

**Figure 6 biosensors-13-00029-f006:**
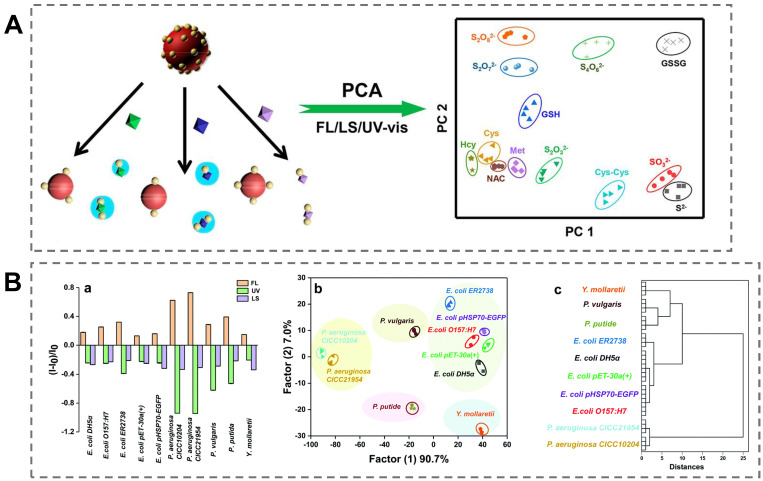
Unsupervised-algorithm-based colorimetric sensor array for pathogen identification. (**A**) A colorimetric sensor array based on the (+)AuNPs/FA-AuAgNCs composite for discrimination of sulfur-containing species and bacteria with PCA algorithm. Reproduced with permission [110]. Copyright 2019, American Chemical Society. (**B**) A colorimetric sensor array based on the (+)AuNPs/AuNCs complex for Gram-negative bacteria identification with HCA algorithm. (**a**) The triple optical response pattern of nanocomplex after incubtion with Gram-negative bacteria. (**b**) LDA for the discrimination of ten different Gram-negative bacteria. (**c**) Dendrogram of HCA for ten different Gram-negative bacteria. Reproduced with permission [111]. Copyright 2022, Royal Society of Chemistry.

**Figure 7 biosensors-13-00029-f007:**
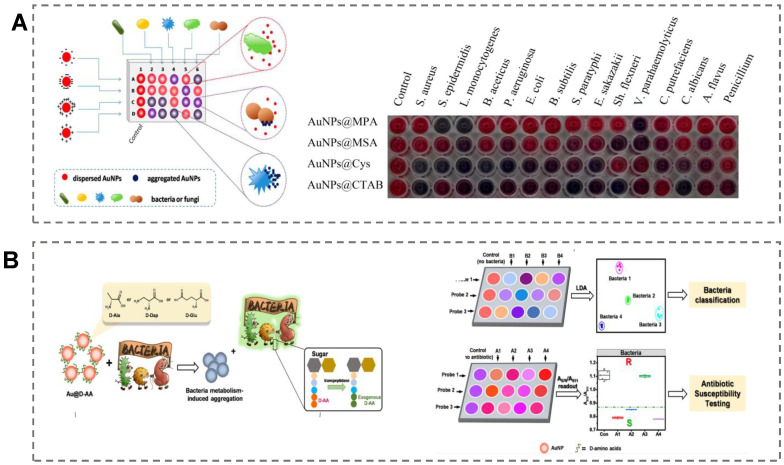
Supervised algorithm-based colorimetric sensor array for pathogen identification. (**A**) A colorimetric sensor array with diverse surface charges of AuNPs for microorganism identification by LDA algorithm. Reproduced with permission [117]. Copyright 2017, American Chemical Society. (**B**) A colorimetric sensor array based on bacteria-metabolism-triggered consumption of D-amino acid–AuNPs for bacteria identification by LDA algorithm. Reproduced with permission [118]. Copyright 2022, American Chemical Society.

**Table 1 biosensors-13-00029-t001:** Applications of colorimetric sensors based on AuNPs for pathogen detection.

Type of Pathogens	Analyte	Functionalization of AuNPs and Strategy	LOD	Ref.
Bacteria	*Lactobacillus* *S. aureus*	Antibody–AuNPs Aggregation	105 CFU/mL 120 CFU/mL	[50]
Bacteria	*E. coli* *S. typhimurium*	Antibody–AuNPs Aggregation	100 CFU/mL	[51]
Virus	SARS-CoV-2	Antibody–AuNPs Aggregation	1000 particles/mL	[52]
Fungi	*A. niger*	Peptide–AuNPs Aggregation	50 CFU/mL	[56]
Bacterial toxins	LPSs	Peptide–AuNPs Aggregation	2.0 nM	[60]
Bacteria	*MRSA*	Aptamer–AuNPs Aggregation	1 CFU/mL	[64]
Bacteria	*E. coli K88*	Aptamer–AuNPs Aggregation	135 CFU/mL	[65]
Virus	SARS-CoV-2 spike protein	Aptamer–AuNPs Aggregation	16 nM	[39]
Bacteria	*S. aureus* MRSA	D-AA–AuNPs Aggregation	10^5^ CFU/mL	[70]
Bacteria	Six types of bacteria	Phages–AuNPs Aggregation	100 CFU/mL	[71]
Bacteria	*S. aureus*	Aptamer–AuNPs Catalytic activity	10 CFU/mL	[75]
Bacteria	*C. jejuni*	Aptamer-Au@Pd NPs Catalytic activity	100 CFU/mL	[76]
Bacteria	*E. coli O157:H7*	Aptamer-Au@Pt NPs Catalytic activity	7 CFU/mL	[77]
Bacteria	*E. coli O157:H7*	PMBA–AuNPs LFA	10^3^ CFU/mL	[86]
Bacteria	*S. enteritidis*	(+)AuNPs LFA	10^3^ CFU/mL	[87]
Virus	SARS-CoV-2	Antibody–AuNPs LFA	-	[89]
Bacteria	*E. coli O157:H7*	antibody–AuNPs Paper chip	4.0 CFU/mL	[92]
Bacteria	*E. coli O157:H7*	AuNPs Microfluidic chip	50 CFU/mL	[96].
Bacteria	*E. coli O157:H7* *S. Typhimurium*	Aptamer–AuNPs Microfluidic chip	1000 CFU/mL 100 CFU/mL	[97]
Bacteria	Five types of bacteria	CTAB–AuNPs sensor array	OD_600_ = 0.005	[110]
Bacteria	Ten types of G^-^ bacteria	CTAB–AuNPs sensor array	OD_600_ = 0.015	[111]
Bacteria	Fifteen types of bacteria	AuNPs with diverse surface charges sensor array	OD_600_ = 0.05	[117]
Bacteria	Eight types of bacteria	DAA–AuNPs sensor array	OD_600_ = 0.1	[118]
Bacteria and fungi	Nineteen types of microorganisms	Biosynthesis of AuNPs sensor array	OD_600_ = 1.0	[119]

## Data Availability

Not applicable.

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
