# Peer review of "Recent Advances in Colorimetric Sensors Based on Gold Nanoparticles for Pathogen Detection"

_biosensors, 2022, doi:10.3390/bios13010029_

Round 1

Reviewer 1 Report

The author reports a review on Recent advances on colorimetric sensor based on gold nanoparticles for pathogen detection. In this review the data collection and analysis have done well for gold nanoparticles as a colorimetric sensor in the application of pathogen detection.  I recommend its publication with minor revision and re review as listed below.

Dear editor,

1. Abstract needs more quantitative and important information.

2. English needs more improvement (I found many grammatical and typographical errors throughout the article; the author should rectify them).

3. Author should improve the literature review using recent reports of gold nanoparticles as a colorimetric sensor in the application of pathogen detection.

4. Data collection from previous reports is well, but needs little detail explanation.

5. Recheck all the abbreviations, figures with their captions.

6. Author should improve the resolution of all the figures.

7. Author should compare gold nanoparticles as a colorimetric sensor with electrochemical sensors and this information must be representing with the following literatures, Anal. Bioanal. Electrochem 10 (4), 488-98, 2018

8. Perspective section must be added before conclusion section.

9. Permission for figures is required.

10. In line 24, the keywords first letter should be capital in all the cases.

11. Information related to gold nanoparticles needs more points as colorimetric sensor in the introduction part.

Reviewer 2 Report

In the review proposed by authors several studies on the employment of gold nanoparticles (AuNPs) for pathogen detection are analyzed and discussed. The authors provide a wide overview of the topic reporting different fabrication strategies and applications in the detection of various pathogenic bacteria. The review is well written and the cited literature is up to date.

Some comments and suggestions for authors are reported below:

1. A comprehensive table summarizing i) the main features of the AuNPs employed in the studies (in terms of AuNPs size, AuNPs capping or surface functionalization); ii) experimental methods; iii) limit of detection; iv) type of target pathogen would be useful to compare the different studies reported.

2. One of the recent approaches against multidrug resistant bacteria is based on the development of nanozymes that are obtained by conjugating active compounds or proteins with nanomaterials such as AuNPs, see for example: /10.1039/D2QM00511E; 10.1016/j.jcis.2020.07.006; 10.3389/fmicb.2018.01441.
This aspect should be mentioned to enrich the list of possible strategies reported in the review.

Reviewer 3 Report

This manuscript gives a review on recent advances on AuNPs-based colorimetric biosensors for pathogen detection. The introduction on machine learning-assisted colorimetric sensor arrays is highlight of the manuscript. Overall, this is a well written review paper that is worth of publication in Biosensors. However, some concerns need to be addressed during the revision.

1. It is of great significance to detect pathogens using AuNPs-based biosensors. In addition to SARS-CoV-2 mentioned in the manuscript, recently monkeypox virus is also a concern for public health. Is there any example that utilizes AuNPs biosensors to realize rapid detection? Please add corresponding information if needed. Also, please check the expression on the virus (E.g., “SARS-CoV-2 virus” not “COVID virus”, Page7, Line 334, while COVID is the name of the disease).

2. The mechanism introduction of AuNPs-based colorimetric effect is too brief. Please add more sentences in the introduction part with more references.

3. Beyond single color change, there are some more advanced biosensing platforms demonstrating the logic gate function based on AuNP-aggregation induced color change (E.g., Sensors and Actuators B: Chemical, 2018, 273(10), Pages 559-565). Please add examples and references.

4. Some labels in the figures are too small to read (E.g., Figure 5c). Please avoid making busy figures with small sizes. Or please increase the size of the labels.
